# Thoughts Are All Over the Place:
# On the Underthinking of Long Reasoning Models

**Yue Wang**[1,*]   **Qiuzhi Liu**[2,*]   **Jiahao Xu**[2,*]   **Tian Liang**[2,*]   **Xingyu Chen**[2,3,*]   **Zhiwei He**[2,3,*]
**Linfeng Song**[2]   **Dian Yu**[2]   **Juntao Li**[1]   **Zhuosheng Zhang**[3]   **Rui Wang**[3]
**Zhaopeng Tu**[2,†]   **Haitao Mi**[2]   **Dong Yu**[2]

[1]Soochow University   [2]Tencent   [3]Shanghai Jiao Tong University

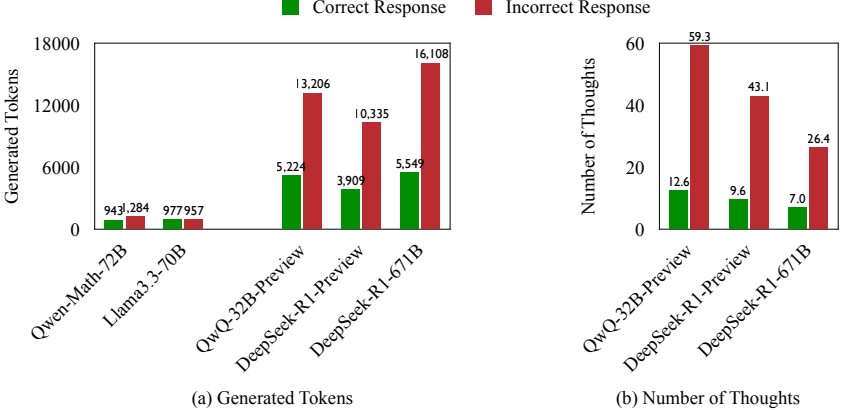

Figure 1: Illustration of the **underthinking issue** on the challenging AIME2024 testset: In LRMs (e.g., QwQ-32B-Preview and DeepSeek-R1-671B), incorrect answers often switch reasoning strategies more frequently than correct ones (Figure b), leading to longer responses without improved accuracy (Figure a). In contrast, conventional LLMs (e.g., Qwen-Math-72B and Llama3.3-70B) show no significant difference in response length between incorrect and correct answers.

## Abstract

Long reasoning models (LRMs) such as OpenAI's o1 and DeepSeek's R1 have demonstrated remarkable abilities in complex reasoning tasks by scaling test-time compute and exhibiting human-like deep thinking. However, we identify a phenomenon we term **underthinking**, where LRMs frequently switch between different reasoning thoughts without sufficiently exploring promising paths to reach a correct solution. This behavior leads to inadequate depth of reasoning and decreased performance, particularly on challenging mathematical problems. To systematically analyze this issue, we conduct experiments on three challenging test sets and two representative open-source LRMs, revealing that frequent thought switching correlates with incorrect responses. We introduce a novel metric to quantify underthinking by measuring token efficiency in incorrect answers. To address underthinking, we propose a decoding strategy with thought switching penalty (TIP) that discourages premature transitions between thoughts, encouraging deeper exploration of each reasoning path. Experimental results demonstrate that our approach improves accuracy across challenging datasets without requiring model fine-tuning.

*Equal Contribution. The work was done when Yue, Xingyu and Zhiwei were interning at Tencent.
†Correspondence to: Zhaopeng Tu <zptu@tencent.com>

39th Conference on Neural Information Processing Systems (NeurIPS 2025).

Our findings contribute to understanding reasoning inefficiencies in LRMs and offer a practical solution to enhance their problem-solving capabilities. Our code is open-source and available at `https://github.com/wangyuenlp/underthinking`.

# 1 Introduction

Long reasoning models (LRMs) (OpenAI, 2024; DeepSeek, 2025; Qwen, 2024; Kimi, 2025) have revolutionized artificial intelligence by enabling models to tackle increasingly complex tasks. LRMs, known for their deep reasoning capabilities, exemplify the potential of large language models (LLMs) to exhibit human-like deep thinking by scaling test-time computation during problem-solving. These models aim to explore diverse reasoning strategies, reflect on their decisions, and iteratively refine solutions, closely mimicking human cognitive processes.

Despite their successes, a critical yet underexplored question remains: **Are long reasoning models thinking deeply enough?** This study provides an initial exploration of this problem. In this work, we investigate a phenomenon we term **underthinking**, which refers to the tendency of LRMs to prematurely abandon promising lines of reasoning, leading to inadequate depth of thought. To systematically analyze underthinking, we conduct experiments on three challenging test sets (e.g., MATH500, GPQA Diamond, and AIME2024) and two open-source LRMs with visible long chains of thought (e.g., QwQ-32B-Preview and DeepSeek-R1-671B). Through extensive analyses, we found that underthinking manifests in the following patterns: (1) it occurs more frequently on harder problems, (2) it leads to frequent switching between different thoughts without reaching a conclusion in each, and (3) it correlates with incorrect responses due to insufficient exploration of reasoning paths. For example, Figure 1 compares the token usage and number of thoughts of correct and incorrect responses. On average, LRMs consume 225% more tokens in incorrect responses than in correct ones due to 418% more frequent thought-switching behaviors.

To quantify this phenomenon, we introduce a novel *underthinking metric* that measures token efficiency in incorrect responses by evaluating the proportion of the response that contributes to reaching correct thoughts. Combining the widely-used accuracy metric with the proposed underthinking metric provides a more comprehensive assessment of LRMs models: accuracy measures how often the model can produce *correct responses*, while the underthinking metric evaluates the token efficiency within *incorrect responses* that contributes to reaching correct thoughts.

In response to these findings, we propose a decoding strategy with thought switching penalty (TIP) that discourages premature transitions between thoughts during the generation process. By adjusting decoding penalties for tokens associated with thought switching, the model is encouraged to thoroughly develop each line of reasoning before considering alternatives. Experimental results show that employing TIP improves accuracy across challenging test sets without requiring additional model fine-tuning.

Our study makes the following contributions:

1. We formally define and characterize the underthinking issue in long reasoning models, where models frequently abandon promising reasoning paths prematurely, leading to inadequate depth of reasoning on challenging problems.

2. We introduce a novel metric to evaluate underthinking by measuring token efficiency in incorrect responses, providing a quantitative framework to assess reasoning inefficiencies.

3. We propose a decoding approach with thought switching penalty (TIP) that encourages models to deeply explore each reasoning thought before switching, improving accuracy without additional model fine-tuning.

# 2 Observing Underthinking Issues

In this section, we present a comprehensive analysis of outputs from LRMs on *challenging math problems*. We begin by illustrating the frequent thinking switch phenomenon observed in responses to these problems, as shown in Figure 2, highlighting how this behavior differs significantly between correct and incorrect answers (Section 2.1). We then show that this phenomenon leads to an inadequate depth of reasoning, causing models to *abandon promising reasoning paths prematurely* (Section 2.2).

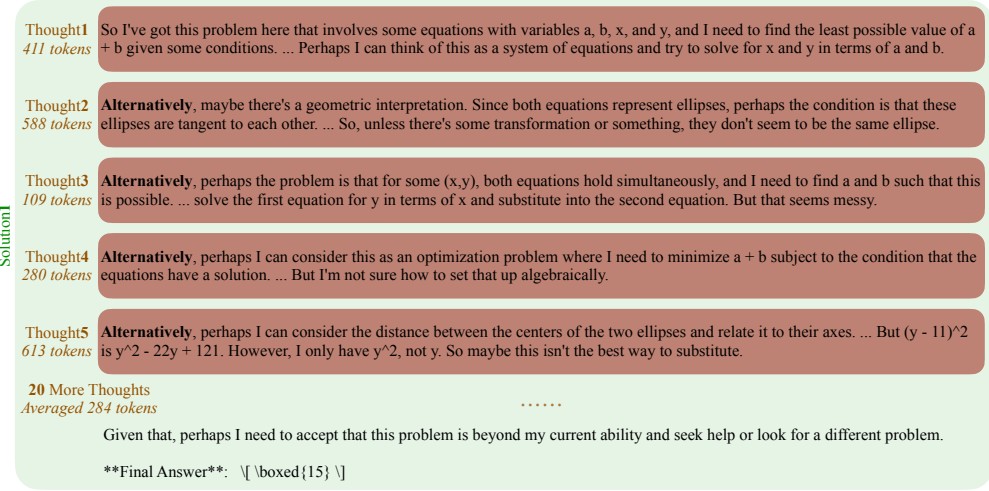

Q: *Let a, b, x, and y be real numbers with a>4 and b>1 such that \frac{x^2}{a^2}+\frac{y^2}{a^2-16} =\frac{(x-20)^2}{b^2-1}+\frac{(y-11)^2}{b^2}=1. Find the least possible value of a+b.*

Solution1

Thought1 *411 tokens* — So I've got this problem here that involves some equations with variables a, b, x, and y, and I need to find the least possible value of a + b given some conditions. ... Perhaps I can think of this as a system of equations and try to solve for x and y in terms of a and b.

Thought2 *588 tokens* — **Alternatively**, maybe there's a geometric interpretation. Since both equations represent ellipses, perhaps the condition is that these ellipses are tangent to each other. ... So, unless there's some transformation or something, they don't seem to be the same ellipse.

Thought3 *109 tokens* — **Alternatively**, perhaps the problem is that for some (x,y), both equations hold simultaneously, and I need to find a and b such that this is possible. ... solve the first equation for y in terms of x and substitute into the second equation. But that seems messy.

Thought4 *280 tokens* — **Alternatively**, perhaps I can consider this as an optimization problem where I need to minimize a + b subject to the condition that the equations have a solution. ... But I'm not sure how to set that up algebraically.

Thought5 *613 tokens* — **Alternatively**, perhaps I can consider the distance between the centers of the two ellipses and relate it to their axes. ... But (y - 11)^2 is y^2 - 22y + 121. However, I only have y^2, not y. So maybe this isn't the best way to substitute.

**20** More Thoughts
*Averaged 284 tokens*
......

Given that, perhaps I need to accept that this problem is beyond my current ability and seek help or look for a different problem.

**Final Answer**: \[ \boxed{15} \]

Figure 2: An example of underthinking issue for QwQ-32B-Preview's output that consists of 25 reasoning thoughts within a single solution. The final correct answer for this example is 23. Thoughts #1,3,4,5 are promising and can lead to the correct answer if explored sufficiently. As intermediate cognitive steps, reasoning thoughts typically does not engage in thorough deduction to get a answer.

Based on this observation, we propose a metric to empirically assess the underthinking issues and present empirical results in Section 2.3. We conclude that *LRMs often underthink when they fail to tackle challenging math problems.*

## 2.1 Frequent Thinking Switch of LRMs

We conduct experiments on three widely-used challenging testsets: **MATH500** (Hendrycks et al., 2021), **GPQA Diamond** (Rein et al., 2023), and **AIME 2022-2024** (MAA Committees). We mainly investigate two widely recognized open-source LRMs featuring visible long CoT: QwQ-32B-Preview and DeepSeek-R1-671B. We also include DeepSeek-R1-Preview to show the development of R1 series models. Given DeepSeek-R1-Preview's daily message limit of 50 via web interface, we evaluated this model solely on the MATH500 and AIME test sets.

**Definition of Reasoning Thoughts**    In this paper, we define *thoughts* as the intermediate cognitive steps within a reasoning solution produced by the model. LRMs often switch reasoning thoughts using terms like "alternatively". For instance, as shown in Figure 2, the problem-solving process involves multiple reasoning thoughts, shifting from algebraic manipulation to geometric interpretation and optimization strategies. The ability to switch between different reasoning strategies allows for a broader exploration of potential solutions and demonstrates the flexibility of the model in tackling complex problems. In this study, we provide a comprehensive analysis of the side effects associated with this ability to switch reasoning thoughts.

We utilize the Llama-3.3-70B model to automatically segment a response into reasoning thoughts due to its superior capabilities in both instruction following and mathematical reasoning. Initially, we manually analyzed responses from the QwQ-32B-Preview model to gather expressions indicative of shifts in thought. We then tasked the Llama-3.3-70B model with scanning the entire response to identify all occurrences of such expressions. Furthermore, we asked the model to determine whether these expressions truly signify a change in thought or merely reflect a stylistic pattern in the response. Only the expressions indicating a genuine thought shift were used as separators for reasoning processes.

**LRMs Switch Thinking More Frequently on Harder Problems**    Figure 3 shows the averaged thoughts and tokens in generated responses across various difficulty levels in the MATH500 test set. Clearly, all models generate more reasoning thoughts with the increase of difficulty level, which is

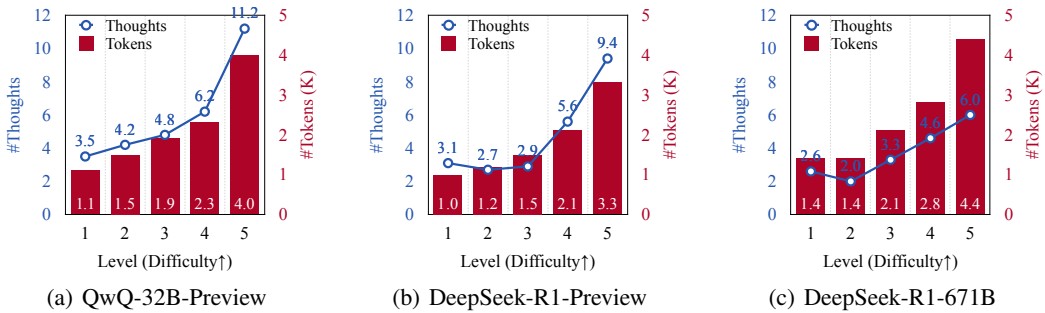

Figure 3: Average number of thoughts and tokens across different difficulty levels on MATH500.

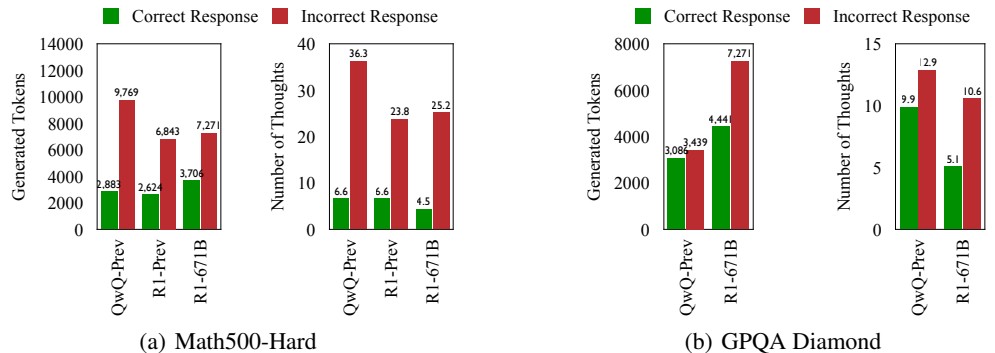

Figure 4: LRMs switch thinking more frequently on incorrect responses, thus expend more tokens without contributing to accuracy.

consistent with the growth of generated tokens. This observation suggests that as the complexity of the problems increases, the models tend to switch thoughts more frequently. This behavior implies that LRMs are able to dynamically adjust their reasoning processes to tackle more challenging problems. The following experiments focus on Level 5 in the MATH500 test set (MATH500-Hard).

**Increased Thought Switching in LRMs during Incorrect Responses**   When examining the behavior of LRMs, we observe a distinct pattern in how they handle incorrect responses. As depicted in Figures 1 and 4, these models exhibit a significant increase in the frequency of thought switching while generating incorrect answers across all test sets. This trend suggests that although the models are designed to dynamically adjust their cognitive processes to solve problems, more frequent thought switching does not necessarily lead to higher accuracy. Essentially, the models may be expending additional computational resources – evidenced by an increase in generated tokens – without achieving more accurate solutions. These insights are crucial because they highlight the need not only to explore additional cognitive pathways when faced with challenges but also to operate in **a more targeted and efficient manner**, thereby improving accuracy even when complex reasoning is required. In the following sections, we empirically validate the inefficiencies associated with frequent thought switching in incorrect responses.

## 2.2   Existence of Underthinking

The behavior of frequent thinking switch in incorrect responses could stem either from (1) genuine underthinking, where the model succeeds in finding promising strategies but fails to stick with them, or from (2) a lack of understanding, prompting it to explore diverse but ineffective approaches. To disentangle these possibilities, we propose an assessment framework that evaluates whether an abandoned reasoning path is actually sufficient to derive a correct answer. By focusing on whether the model can persistently follow and deepen a single, promising line of thought, we can identify instances of underthinking.

**Assessing Thought Correctness**   In the example presented in Figure 2, we observe that some early thoughts may lead to the correct answer. For instance, Thought 1 initiates a correct interpretation

by recognizing that the given equations resemble those of ellipses centered at (0,0) and (20,11). Setting the two expressions equal is a valid approach to finding common points $(x, y)$ that satisfy both equations. Instead of concentrating on thoroughly exploring the plausible thought with further algebraic manipulation and optimization techniques, the model frequently shifts its focus and uses approximately 7,270 additional tokens without arriving at a correct answer. Ultimately, it concludes with a guessed answer that lacks support from the extended COT process.

We leverage LLMs to assess whether each thought leads to a correct answer using the prompt detailed in Appendix A.Specifically, we use two models distilled from DeepSeek-R1-671B based on `Llama` and `Qwen` – *DeepSeek-R1-Distill-Llama-70B* and *DeepSeek-R1-Distill-Qwen-32B*, which achieve new state-of-the-art results for dense models across various reasoning benchmarks. If at least one model generates a confidence score of 2 for a thought, we regard it as a correct thought.

We evaluate the accuracy of our assessment approach using responses generated by Qwen-32B-Preview for 90 instances from the AIME 22-24 test sets. We utilize the final thought in each response as the test example and its correctness as the ground-truth label. To ensure a fair comparison, we randomly streamline correct thoughts to match the average length of incorrect thoughts. Ultimately, we have 35 correct thoughts with an average length of 278.1 tokens and 55 incorrect thoughts with an average length of 278.3 tokens. Our assessment approach achieves accuracies of **82.9%** for correct examples and **81.8%** for incorrect examples, demonstrating its effectiveness.

**Early-Stage Thoughts Are Correct but Abandoned in Incorrect Responses** Figure 5 depicts the ratio of correct thoughts at each index in incorrect responses on the three challenging test sets. The analysis highlights a critical insight into the phenomenon of underthinking. Specifically, a notable proportion of initial thoughts across various models were correct but were not pursued to completion. This tendency to abruptly shift away from these promising thoughts indicates an inadequate depth of reasoning, where potentially correct solutions are prematurely abandoned before being thoroughly explored. This observation suggests a need for enhancing the models' ability to persistently explore a specific line of reasoning deeply and accurately before opting to switch to alternative thought processes.

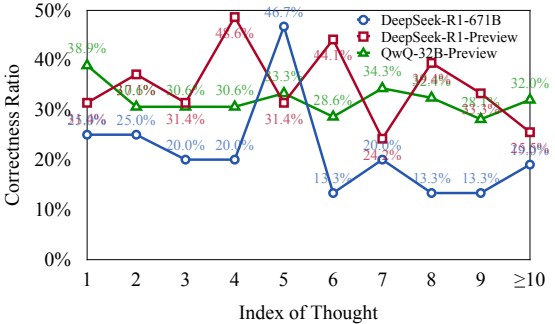

Figure 5: The ratio of correct reasoning thoughts at each index in incorrect responses.

**Most Incorrect Responses Contain Correct Thoughts** Figure 6 plots the thought correctness ratios in incorrect responses from various models. We observe that over 70% of incorrect responses contain at least one correct thought. Furthermore, in more than 50% of these responses, over 10% of the thoughts are correct. Combined with observations from Figure 5, this suggests that while LRMs can initiate correct reasoning pathways, they may struggle to continue these pathways to reach the correct conclusion. This highlights the importance of encouraging models to maintain and expand their **initial correct thoughts** to synthesize them into accurate final answers.

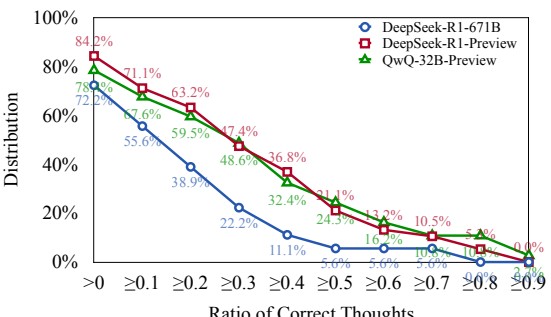

Figure 6: The distribution of thought correctness ratio in incorrect responses.

These insights lead us to propose an underthinking metric based on the presence of the first correct thought in the subsequent section.

## 2.3 Empirical Underthinking Results

In this section, we propose a metric for empirically assessing underthinking issues based on token efficiency, complementing the widely used accuracy metric.

**Underthinking Metric** Intuitively, if a model generates a correct thought at an early stage and then switches to other thoughts without reaching a correct answer, the tokens generated thereafter do not contribute to reaching a correct solution and are considered inefficient due to underthinking. The underthinking score, denoted as $\xi_{UT}$, is defined as:

$$\xi_{UT} = \frac{1}{N} \sum_{i=1}^{N} \left( 1 - \frac{\hat{T}_i}{T_i} \right) \tag{1}$$

Here, $N$ represents the number of instances in a given test set where the evaluated model generates incorrect responses. $T_i$ is the total number of tokens in the $i$-th incorrect response, and $\hat{T}_i$ is the number of tokens from the beginning of that response up to and including the first correct thought. If there is no correct thought in the $i$-th response, $\hat{T}_i = T_i$, indicating that the model lacks an understanding of this problem, leading it to explore diverse but ineffective approaches. Therefore, it cannot be considered underthinking. Consider Figure 2 as an example: the first reasoning thought can reach a correct answer if fully explored, with $\hat{T} = 411$. Consequently, $\xi_{UT} = 1 - \frac{411}{7681} = 0.946$, which can be considered extremely inefficient, reflecting a high underthinking score.

The metric $\xi_{UT}$ complements the accuracy metric by quantifying the extent of underthinking by measuring the token efficiency in generating effective content within **an incorrect response**:

- A lower value of $\xi_{UT}$ indicates higher token efficiency, meaning that a greater proportion of tokens in incorrect responses contribute towards reaching a correct thought before switching to another thought. This suggests that the model is more efficient in its token utilization even when it fails to provide a correct answer.

- A higher value of $\xi_{UT}$ signifies lower token efficiency, indicating that a larger number of tokens do not contribute effectively towards generating a correct thought. This reflects greater underthinking, where the model generates redundant or irrelevant tokens by frequently switching thoughts.

**Empirical Results** Table 1 provides insights into model performance across challenging test sets, evaluating both accuracy and underthinking (UT) scores. Clearly, all LRMs suffer from significant underthinking issues, although there are considerable differences across models and test sets. The results reveals that the relationship between model accuracy and underthinking varies across different datasets. On the MATH500-Hard and GPQA Diamond datasets, higher accuracy achieved by the superior DeepSeek-R1-671B model is accompanied by higher UT Scores, indicating more underthinking in incorrect responses.

Table 1: Underthinking scores on challenging testsets.

| Models | Accuracy($\uparrow$) | UT Score ($\downarrow$) |
|---|---|---|
| *MATH500-Hard (Level 5)* | | |
| QwQ-32B-Preview | 84.3 | 58.2 |
| DeepSeek-R1-Preview | 83.6 | 61.5 |
| DeepSeek-R1-671B | 92.5 | 65.4 |
| *GPQA Diamond* | | |
| QwQ-32B-Preview | 59.6 | 48.3 |
| DeepSeek-R1-671B | 73.2 | 58.8 |
| *AIME2024* | | |
| QwQ-32B-Preview | 46.7 | 65.0 |
| DeepSeek-R1-Preview | 46.7 | 75.7 |
| DeepSeek-R1-671B | 73.3 | 37.0 |

This suggests that while the model is more capable overall, it may produce longer but less effective reasoning when uncertain, possibly due to exploring multiple incorrect reasoning paths without efficiently converging on the correct solution. Conversely, on the AIME2024 test set, the DeepSeek-R1-671B model not only attains higher accuracy but also exhibits a lower UT score, reflecting less underthinking and greater token efficiency. This implies that the model's reasoning remains focused and effective even when it does not arrive at the correct answer, perhaps due to better alignment with the problem types and reasoning processes required by the AIME2024 task.

These findings illustrate that underthinking behavior is sensitive to the nature of the dataset and the tasks involved. The larger model's superior capabilities do not uniformly translate to less

Table 2: Underthinking scores of Qwen3 Family on AIME24. We report the average number of thought-switching tokens ($\widehat{V}$ in Equation 3.1) and the average interval between them.

| Models | Accuracy (↑) | | | | Switching Tokens | | Weighted |
|---|---|---|---|---|---|---|---|
| | Pass@1 | Pass@4 | Pass@8 | Pass@16 | Number | Interval | UT Score (↓) |
| Qwen3-4B | 65.6 | 79.5 | 81.6 | 83.3 | 27.6 | 372.1 | $15.0_{\pm 18.5}$ |
| Qwen3-8B | 64.6 | 78.0 | 81.6 | 83.3 | 20.3 | 561.4 | $16.8_{\pm 20.6}$ |
| Qwen3-14B | 70.8 | 82.3 | 86.6 | 90.0 | 18.2 | 506.6 | $13.4_{\pm 20.1}$ |
| Qwen3-32B | 73.8 | 87.5 | 90.0 | 93.3 | 13.7 | 681.3 | $10.6_{\pm 14.5}$ |

underthinking across all tasks. In some cases, increased model capacity leads to more elaborate but inefficient reasoning in incorrect responses, while in others, it enhances both accuracy and reasoning efficiency. Understanding the underthinking phenomenon is crucial for developing models that not only provide correct answers but also exhibit effective reasoning processes.

Furthermore, due to different training strategies, models across different families may differ fundamentally: some reach correctness intuitively (low UT score), while others do so through iterative refinement (higher UT score). Since both approaches can achieve high accuracy, this observation explains the weak correlation between UT score and accuracy shown in Table 1. Nevertheless, within specific model families, UT scores show a clearer correlation with accuracy. This trend may result from similar training strategies that reduce behavioral variance, as supported by the results in Table 2. Within the Qwen3 family, for instance, we observed a clear trend where the severity of underthinking decreases as model size increases.

## 3 Mitigating Underthinking Issues

In this section, we propose a lightweight mechanism that mitigates underthinking issues without requiring any model fine-tuning.

### 3.1 Decoding with Thought Switching Penalty

Aforementioned findings show that LRMs prioritize exploring many solutions over deeply investigating one. Inspired by the success of the coverage penalty in neural machine translation (Tu et al., 2016; Wu et al., 2016), we propose a novel decoding algorithm with a *thought switching penalty* to encourage the model to explore potential thoughts more thoroughly before moving on to new ones.

**Standard Decoding** In standard decoding, the probability of each token $v$ at position $t$ is computed using the softmax function over the logits $\mathbf{z}_t \in \mathbb{R}^{|V|}$ ($|V|$ is the vocabulary size) in the output layer:

$$P(x_t = v | x_{<t}) = \frac{\exp(z_{t,v})}{\sum_{v' \in V} \exp(z_{t,v'})}$$

where $z_{t,v} \in \mathbf{z}_t$ is the logit for token $v$. By repeating this step for each position in the sequence, the model generates sequences of tokens, computing probabilities for each possible continuation.

**Thought Switching Penalty (TIP)** To encourage the model to delve deeper into current thoughts before switching, we introduce a penalty on tokens that are associated with thought transitions. A key consideration in designing this penalty is to distinguish between unproductive, rapid thought-switching and deliberate, strategic shifts in reasoning (e.g., backtracking) (Gandhi et al., 2025). Therefore, TIP is designed to be selective: it specifically targets and penalizes thought-switching tokens only when they appear with high frequency within a recent context, a pattern indicative of shallow exploration. This ensures that isolated or intentional shifts in thought remain unpenalized, preserving the model's ability to employ effective complex reasoning strategies like backward chaining. Let $\widehat{V} \subset V$ be the set of tokens associated with thought switching (e.g., "alternatively"). We modify the logits as follows:

$$\hat{z}_{t,v} = \begin{cases} z_{t,v} - \alpha, & \text{if } v \in \widehat{V} \text{ and } t < \Psi + \beta \\ z_{t,v}, & \text{otherwise} \end{cases}$$

Table 3: Pass@k performance of the proposed TIP method. We report the average number of thought-switching tokens ($\hat{V}$ in Equation 3.1) and the average interval between them in the generated samples.

| Models | Accuracy (↑) | | | | Switching Tokens | | Weighted |
|---|---|---|---|---|---|---|---|
| | Pass@1 | Pass@4 | Pass@8 | Pass@16 | Number | Interval | UT Score (↓) |
| *MATH500-Hard (Level 5)* | | | | | | | |
| QwQ-32B-Preview | 83.1 | 92.4 | 94.4 | 95.8 | 12.6 | 445.6 | $11.7_{\pm 20.5}$ |
| + TIP | 83.7 | 93.2 | 95.3 | 96.4 | 5.7 | 517.6 | $11.0_{\pm 19.5}$ |
| R1-Distill-Qwen-32B | 88.3 | 94.5 | 96.1 | 97.0 | 6.7 | 792.5 | $3.3_{\pm 8.8}$ |
| + TIP | 89.4 | 94.6 | 96.1 | 97.0 | 2.7 | 964.0 | $3.0_{\pm 8.5}$ |
| *GPQA Diamond* | | | | | | | |
| QwQ-32B-Preview | 57.6 | 78.5 | 85.3 | 90.3 | 21.1 | 356.8 | $25.1_{\pm 23.9}$ |
| + TIP | 59.1 | 78.9 | 85.8 | 91.2 | 7.3 | 432.5 | $23.2_{\pm 23.2}$ |
| R1-Distill-Qwen-32B | 61.6 | 78.1 | 83.6 | 86.9 | 13.4 | 548.6 | $22.3_{\pm 25.0}$ |
| + TIP | 61.7 | 80.2 | 86.6 | 90.4 | 4.6 | 747.1 | $23.1_{\pm 25.3}$ |
| *AIME2024* | | | | | | | |
| QwQ-32B-Preview | 38.3 | 53.7 | 58.5 | 62.7 | 16.1 | 459.7 | $40.6_{\pm 28.4}$ |
| + TIP | 44.1 | 61.6 | 68.3 | 74.0 | 13.9 | 515.7 | $35.8_{\pm 27.8}$ |
| R1-Distill-Qwen-32B | 61.4 | 75.9 | 79.1 | 81.7 | 8.2 | 819.5 | $19.6_{\pm 20.6}$ |
| + TIP | 64.1 | 79.0 | 81.7 | 83.0 | 4.5 | 1018.0 | $17.7_{\pm 20.6}$ |
| DeepSeek-R1 | 73.8 | 86.2 | 88.8 | 89.8 | 13.8 | 580.1 | $14.6_{\pm 19.1}$ |
| + TIP | 74.8 | 86.4 | 88.8 | 89.8 | 5.7 | 941.6 | $13.0_{\pm 18.0}$ |

where

- $\alpha \geq 0$ (*Penalty Strength*) is a parameter controlling the strength of the penalty applied to thought-switching tokens. A larger $\alpha$ results in a greater reduction of the logits for these tokens, making them less likely to be chosen.

- $\beta \geq 0$ (*Penalty Duration*) specifies the number of positions from the start of a thought at $\Psi$, during which the penalty is active. A larger $\beta$ extends the penalty over more positions, further discouraging early thought switching.

When $\alpha = 0$ or $\beta = 0$, the penalty is effectively disabled, and the decoding process reduces to the standard decoding algorithm. The adjusted logits $\hat{z}_{t,v}$ reduce the probability of generating thought-switching tokens within a specified window, encouraging the model to continue expanding on the current thought before moving on. The new probability distribution becomes:

$$\hat{P}(x_t = v \mid x_{<t}) = \frac{\exp(\hat{z}_{t,v})}{\sum_{v' \in V} \exp(\hat{z}_{t,v'})}$$

### 3.2 Experimental Results

For reliable evaluation, we report Pass@1 computed from 32 samples per problem with a temperature of 0.7 and a top_p value of 0.95. We tuned $\alpha \in \{3, 5, 10, 20, 30\}$ and $\beta \in \{300, 400, 500, 600, 700\}$ on the AIME 2022-2023 development set using QwQ-32B-Preview, selecting the best pair $\alpha = 3, \beta = 600$ for all models and benchmarks. Please refer Appendix B.2 for more details.

**Standard Decoding** Table 3 shows that our TIP method consistently improves performance in all cases by mitigating the underthinking issues. On AIME-24, Pass@1 improves by 5.8% on QwQ-32B-Preview, 2.7% on R1-Distill-Qwen, and 1.0% on DeepSeek-R1, while the underthinking (UT) score drops across the board. Observing the indicators, we see fewer thought-switching tokens and larger intervals when using TIP, confirming that our method encourages models to explore individual reasoning paths more thoroughly and mitigates underthinking. For example, when applying TIP to DeepSeek-R1 on AIME2024, the average thought-switching tokens decreased (13.8→5.7), and the average interval between switches increased (580.1→941.6). These changes reflect fewer premature transitions, resulting in a more focused and human-like exploration of reasoning paths.

Table 4: Results of TIP for best-of-N sampling on AIME2024. We conducted 10,000 trials by randomly sampling from the 32 samples and reported the average results. "(Averaged)" denotes the average performance over 32 samples.

| Models | 4 Samples | | 8 Samples | | 16 Samples | |
|---|---|---|---|---|---|---|
| | Acc.($\uparrow$) | UT ($\downarrow$) | Acc.($\uparrow$) | UT ($\downarrow$) | Acc.($\uparrow$) | UT ($\downarrow$) |
| QwQ (Averaged) | 38.4 | 40.5 | 38.3 | 40.6 | 38.3 | 40.6 |
| + TIP (Averaged) | 44.1 | 35.8 | 44.0 | 35.9 | 44.0 | 35.9 |
| QwQ + Self-Consistency | 43.7 | 35.4 | 44.3 | 34.0 | 44.6 | 31.9 |
| + TIP | 51.4 | 26.6 | 53.4 | 24.3 | 53.9 | 24.1 |
| QwQ + Laconic Decoding | 47.0 | 28.2 | 47.0 | 25.5 | 45.1 | 24.0 |
| + TIP | 50.3 | 26.7 | 51.6 | 23.3 | 50.9 | 20.8 |
| R1-Distill-Qwen (Averaged) | 61.4 | 19.2 | 61.3 | 19.2 | 61.3 | 19.1 |
| + TIP (Averaged) | 64.1 | 17.8 | 64.0 | 17.7 | 64.1 | 17.7 |
| R1-Distill-Qwen + Self-Consistency | 67.0 | 13.4 | 67.8 | 11.4 | 68.9 | 8.9 |
| + TIP | 69.9 | 12.5 | 71.4 | 11.0 | 72.3 | 9.1 |
| R1-Distill-Qwen + Laconic Decoding | 71.1 | 11.3 | 74.4 | 8.7 | 77.5 | 7.4 |
| + TIP | 75.4 | 9.8 | 78.0 | 7.3 | 77.9 | 6.5 |
| R1 (Averaged) | 73.9 | 14.5 | 73.7 | 14.6 | 73.8 | 14.5 |
| + TIP (Averaged) | 74.8 | 13.0 | 74.8 | 12.9 | 74.8 | 13.0 |
| R1 + Self-Consistency | 79.3 | 10.1 | 79.8 | 9.8 | 79.7 | 9.5 |
| + TIP | 81.3 | 7.5 | 82.2 | 6.4 | 82.1 | 5.8 |
| R1 + Laconic Decoding | 81.4 | 8.1 | 82.6 | 6.2 | 83.2 | 5.1 |
| + TIP | 83.1 | 7.4 | 83.8 | 6.6 | 83.3 | 6.7 |

**Best-of-N Sampling** To further assess TIP, we combined it with two widely used best-of-N sampling methods: (1) **Self-Consistency** (Wang et al., 2023), selecting the most consistent answer from multiple samples; (2) **Laconic Decoding** (Raoof & Dimakis, 2025), selecting the shortest of multiple generated answers, based on the observation that correct responses often have fewer tokens.

Table 4 indicates that using TIP consistently boosts accuracy across all model-method combinations. For instance, applying TIP to Self-Consistency with QwQ-32B-Preview (4-sample setting) significantly raises accuracy (43.7%→51.4%) and reduces UT scores (35.4→26.6). Similarly, combining Laconic Decoding with TIP yields consistent gains, particularly pronounced for stronger models (e.g., R1-Distill-Qwen: 74.4%→78.0% at 8 samples). These results clearly demonstrate that the TIP method successfully complements existing sampling strategies, encouraging more thorough reasoning and reliably mitigating underthinking issues in challenging mathematical reasoning scenarios.

Overall, the TIP approach represents a significant step toward addressing the underthinking problem in LRMs. Although the hyperparameters are tuned on the AIME 2022 and 2023 test sets using the QwQ-32B-Preview model, the consistent improvements observed across various test sets and models with the same hyperparameters validate the generalizability of the method. Our findings also suggest that the TIP method synergizes well with best-of-N sampling strategies, leading to further improvements in accuracy and reductions in underthinking scores. This indicates that encouraging more thorough exploration of individual reasoning paths complements the diversity introduced by sampling methods.

## 4 Related Work

**Scaling Test-Time Compute** Recent advancements in deep reasoning models, such as OpenAI's o1, have emphasized scaling test-time compute to improve complex problem-solving capabilities. One approach focuses on **expanding the search space** by considering multiple candidate solutions during decoding, exemplified by self-consistency (Wang et al., 2023), best-of-n decoding, and minimum Bayes risk decoding (Lightman et al., 2024; Li et al., 2023; Khanov et al., 2024; Heineman et al., 2024; Wu et al., 2024). Another influential direction involves promoting **human-like deep thinking**, beginning with Chain-of-Thought (Wei et al., 2022), where models mimic human reasoning

processes (Cesista, 2024; Pfau et al., 2024). Recent models, such as QwQ (Qwen, 2024), DeepSeek-R1 (DeepSeek, 2025), and Kimi-1.5 (Kimi, 2025), leverage reinforcement learning (RL) to enable strategic, reflective reasoning and improve accuracy in complex tasks.

**Efficient Thinking**   Efficient reasoning matters as much for language reasoning models (LRMs) as it does for human cognition. Models sometimes suffer **overthinking**, wasting resources on trivial paths (Chen et al., 2024). In contrast, we focus on the less-explored issue of **underthinking**, where models prematurely abandon promising reasoning directions, limiting performance on challenging problems. Along this direction, recent research has begun exploring methods to enhance reasoning efficiency. For instance, Laconic decoding implements shortest-of-n decoding strategies to minimize error-prone lengthy responses (Raoof & Dimakis, 2025), while Muennighoff et al. (2025) propose techniques to optimize test-time computation through dynamic termination signals.

**Manipulating Decoding Penalties**   Penalty mechanisms in NLP decoding have become increasingly relevant to addressing shortcomings in traditional search methods. Length normalization penalties (Jean et al., 2015; Koehn & Knowles, 2017; Tu et al., 2017; Murray & Chiang, 2018) encourage appropriate translation lengths, improving fluency and adequacy. Additionally, coverage penalties (Tu et al., 2016; Wu et al., 2016) reduce textual redundancies by ensuring comprehensive attention over source tokens. See et al. (2017) also applied these ideas to summarization tasks. In this work, we introduce specific decoding penalties to mitigate underthinking, prompting models to sustain deeper reflection rather than superficial or frequent strategy shifts. To our knowledge, we are the first to investigate decoding penalties to address the underthinking problem.

## 5   Conclusion

In this work, we identified underthinking as a key limitation for LRMs performing challenging reasoning tasks, highlighting how premature abandonment of promising paths reduces efficiency and accuracy. We developed a new metric to quantify underthinking based on token efficiency in incorrect responses, and introduced a decoding strategy – thought switching penalty (TIP) – to encourage deeper exploration before shifting reasoning directions. Empirical evaluations show that TIP significantly reduces underthinking and improves performance on difficult reasoning benchmarks, without requiring additional model training.

This work contributes to a deeper understanding of reasoning processes in LRMs and provides a practical approach to align their problem-solving capabilities. Future directions include exploring adaptive mechanisms within models to self-regulate thought transitions and further improving reasoning efficiency.

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

# A  Assessing Thought Correctness

In the example presented in Figure 2, we observe that some early thoughts may lead to the correct answer. For instance, Thought 1 initiates a correct interpretation by recognizing that the given equations resemble those of ellipses centered at (0,0) and (20,11). Setting the two expressions equal is a valid approach to finding common points $(x, y)$ that satisfy both equations. Instead of concentrating on thoroughly exploring the plausible thought with further algebraic manipulation and optimization techniques, the model frequently shifts its focus and uses approximately 7,270 additional tokens without arriving at a correct answer. Ultimately, it concludes with a guessed answer that lacks support from the extended COT process.

We leverage LLMs to assess whether each thought leads to a correct answer using the following prompt:

---

Problem P = {problem}
Solution Draft S = {split solutions}
Correct Answer A = {expected answer}

1. *Please analyze the relevance between the solution S and the problem P, and conduct some verifications to check the correctness of the solution itself. Please think step by step to give an explanation **EXPLANATION**.*
2. *If you think the solution draft S can lead to the correct answer A of the problem P, please stick to the line of thinking without deviation and carry it through to completion. If you think it cannot yield the correct answer or you're not sure, don't force yourself to give an answer and generate **None**.*
3. *Please tell me honestly how confident you are that you can solve the problem P correctly based on the the solution draft S. Out of 2, please generate your confidence score **CONFIDENT_SCORE**.*

Please output **EXPLANATION** and **CONFIDENT_SCORE** according to the following format:
EXPLANATION: \boxed{}
CONFIDENT_SCORE: \boxed{}

---

Specifically, we use two models distilled from DeepSeek-R1-671B based on `Llama` and `Qwen` – *DeepSeek-R1-Distill-Llama-70B* and *DeepSeek-R1-Distill-Qwen-32B*, which achieve new state-of-the-art results for dense models across various reasoning benchmarks. If at least one model generates a confidence score of 2 for a thought, we regard it as a correct thought.

We evaluate the accuracy of our assessment approach using responses generated by Qwen-32B-Preview for 90 instances from the AIME 2022, 2023, and 2024 test sets. We utilize the final thought in each response as the test example and its correctness as the ground-truth label. To ensure a fair comparison, we randomly streamline correct thoughts to match the average length of incorrect thoughts. Ultimately, we have 35 correct thoughts with an average length of 278.1 tokens and 55 incorrect thoughts with an average length of 278.3 tokens. Our assessment approach achieves accuracies of 82.9% for correct examples and 81.8% for incorrect examples, demonstrating its effectiveness.

# B  Experimental Details of TIP

## B.1  Selection of Thought-Switching Tokens

A key component of our TIP decoding strategy is the use of a predefined set of thought-switching tokens that signal a potential switch in the reasoning path. The selection of these tokens is treated as a task-specific hyperparameter. For our experiments, they were chosen empirically based on a qualitative analysis of common thought-switching patterns observed in the model's outputs. Specifically, we select alternative and messy as thought-switching tokens in our implementation. This approach provides a simple yet effective mechanism that can be adapted to different tasks or model behaviors without extensive tuning. Besides, since LLM tokenizers can split a single word into

Table 5: Accuracy on AIME2022-23 with respect to different values of $\alpha$ and $\beta$.

| **Pass@1** | | $\alpha$ | | | |
|---|---|---|---|---|---|
| **Accuracy** | | *3* | *5* | *10* | *20* |
| | 300 | 35.2 | 37.0 | 39.0 | 39.4 |
| | 400 | 39.3 | 37.1 | 37.1 | 38.4 |
| $\beta$ | 500 | 38.5 | 38.7 | 39.1 | 39.2 |
| | 600 | **39.8** | 39.4 | 38.0 | 38.0 |
| | 700 | 37.1 | 39.4 | 39.0 | 38.3 |

multiple subword tokens, the switching penalty is applied exclusively to the first subword token of a designated thought-switching word.

## B.2   Grid Search of $\alpha$ and $\beta$

To ensure robust conclusions, we report Pass@1 results computed from 32 samples per instance. We calculate the weighted underthinking score for each instance over its 32 samples:

$$\xi_{wUT} = \frac{1}{32} \sum_{i=1}^{32} \xi_{UT}(s_i) \tag{2}$$

where $s_i$ is the $i$-th sample of the instance, and $\xi_{UT}(s_i) = 0$ when $s_i$ is correct.

By adjusting $\alpha$ and $\beta$, we can control the model's behavior to achieve the desired level of thought exploration. We performed a grid search with $\alpha$ values in $[3, 5, 10, 20, 30]$ and $\beta$ values in $[300, 400, 500, 600, 700]$ using a development set that included the AIME 2022 and 2023 test sets. Table 5 lists the impact of varying the penalty strength $\alpha$ and penalty duration $\beta$ on the model's accuracy. We observe that increasing the penalty strength $\alpha$ generally leads to an improvement in accuracy up to a certain threshold, after which the benefits plateau or even diminish. Adjusting the penalty duration $\beta$ also significantly affects performance: At a lower penalty strength ($\alpha = 3$), increasing $\beta$ from 300 to 600 results in accuracy gains from 35.2% to 39.8%, the highest observed accuracy in our experiment. Conversely, at higher penalty strengths ($\alpha = 20$), extending $\beta$ beyond 300 leads to a decrease in accuracy, indicating that too long a penalty duration can hinder performance when combined with a strong penalty. We selected $\alpha = 3$ and $\beta = 600$ for our subsequent experiments.

## C   Generalization to Multimodal Reasoning

To investigate if underthinking extends beyond text-only domains, we evaluated two vision-language models, GLM-4.1V-Thinking (Hong et al., 2025) and MiMo-VL-7B-RL (Xiaomi, 2025), on the OE_MM_maths_en_COMP subset of OlympiadBench (He et al., 2024). This benchmark contains competition-level mathematics problems that require synthesizing information from both text and images. As shown in Table 6, both models exhibit underthinking. This finding leads to two implications: first, our framework can be adapted to analyse multimodal reasoning, and second, it suggests that underthinking is a general reasoning failure not confined to a single modality.

Table 6: Underthinking scores on challenging testsets.

| **Models** | **Accuracy**($\uparrow$) | **UT Score** ($\downarrow$) |
|---|---|---|
| MiMo-VL-7B-RL | 56.7 | 4.8 |
| GLM-4.1V-Thinking | 58.7 | 17.1 |

