# OpenReview forum: "Thoughts Are All Over the Place: On the Underthinking of Long Reasoning Models"
_NeurIPS.cc/2025/Conference — NeurIPS 2025 spotlight_

### Official Review · Reviewer_MhYP · 2025-06-30

**Clarity:** 3
**Significance:** 3
**Originality:** 3
**Rating:** 5
**Confidence:** 4

**Summary:**

This paper studies underthinking, a phenomenon where a LRM frequently switches between reasoning thoughts before fully exploring promising paths. The authors found that incorrect responses often have a significantly higher number of thoughts, even though many of those thoughts could have led to correct answers. They further propose a UT score to quantitatively measure this issue. Based on these findings, the paper proposes a decoding strategy TIP that uses a thought-switching penalty. Experiments show that this simple strategy leads to improved performance for both standard decoding and best-of-N sampling while maintaining high token efficiency.

**Questions:**

- The TIP decoding strategy requires a predefined set of thought-switching tokens. How are these tokens selected? What if a word is tokenized into multiple tokens?
- The legend for Figure 5 is confusing. There are dots in the figure that look different from the rest. What is the difference between these dots and the others?

**Ethical Concerns:**

["NO or VERY MINOR ethics concerns only"]

**Final Justification:**

Overall, I think it's a solid paper that studies an important problem in reasoning models. The experimental results could provide some insights for future research. There were some points of confusion in the paper, which were clarified in the rebuttal.

**Limitations:**

The authors briefly discuss the limitations in Section 2.3 (as stated in the checklist). I would suggest that the authors explicitly discuss this in a "Limitations" section to make things clear.

**Quality:**

3

**Strengths And Weaknesses:**

Strengths
- This paper is well-motivated. It tackles an important problem in LRMs and provides insights into model performance and token efficiency.
- The experiments and analysis are clean and easy to follow.
- The proposed strategy, TIP, effectively mitigates the identified issues without significant overhead.

Weakness
- My main concern with this paper is the proposed UT score. If I understand correctly, the score measures the percentage of tokens required to achieve the first correct thought, but not how frequently the model switches between thoughts. It is still possible for a model to quickly reach the first correct thought (therefore having a low UT score), but then switch between different thoughts afterward. Therefore, I’m not sure if the UT score correctly captures what the authors intended to measure.
- Additionally, based on Table 1, there is not a clear correlation between accuracy and the UT score, so it is unclear if the proposed metric can provide useful insights on the relation between underthinking issues and model performance. Instead, I think the "switch tokens" number and the interval measure in Table 2 would be more informative.

---

> ### Author Rebuttal · Authors · 2025-07-30
>
> Thank you for your strong endorsement (rating 5) and for noting our clean motivation, experiments, and TIP's effectiveness in improving performance and efficiency. We appreciate your detailed questions on metrics and implementation — they've helped clarify key aspects.
>
> ---
>
> > Q1: My main concern with this paper is the proposed UT score. If I understand correctly, the score measures the percentage of tokens required to achieve the first correct thought, but not how frequently the model switches between thoughts. It is still possible for a model to quickly reach the first correct thought (therefore having a low UT score), but then switch between different thoughts afterward. Therefore, I’m not sure if the UT score correctly captures what the authors intended to measure.
> >
>
> A1: Thank you for the opportunity to clarify. **The UT score is intentionally designed to measure the *inefficiency* caused by switching *after* a correct path has been found, not just the raw frequency of switching.**
> A model that quickly finds a correct thought (low initial token use) but then switches away from it frequently will receive a high UT score, as many tokens are "wasted" after the first point of success. Conversely, a model that finds a correct thought and develops it to the final answer will have a UT score near zero. This design directly operationalizes our definition of underthinking: prematurely abandoning a promising path. We apologize if this was not perfectly clear and will refine the description in Section 2.3.
>
> ---
>
> > Q2: Additionally, based on Table 1, there is not a clear correlation between accuracy and the UT score, so it is unclear if the proposed metric can provide useful insights on the relation between underthinking issues and model performance. Instead, I think the "switch tokens" number and the interval measure in Table 2 would be more informative.
> >
>
> A2: This is a very insightful observation. You are correct that there isn't a simple, universal correlation *across different model families* in Table 1, and for good reason.
>
> 1. **UT Score vs. "Switch Tokens":** As you note, other metrics like "switch tokens" are informative, and we include them for this reason. However, switching itself is not always bad; abandoning an incorrect path is a feature of good reasoning. The UT score is more nuanced because it specifically penalizes abandoning *correct* paths.
> 2. **Cross-Family vs. Intra-Family Correlation:** Different model families may use fundamentally different strategies (e.g., some are trained for iterative refinement, others for more direct solutions). This can lead to high accuracy with either high or low UT scores, masking a direct correlation. However, as shown in our new results for **Reviewer-Ffc3**, **within a single model family (Qwen3), there is a clear negative correlation between model size/accuracy and the UT score.** This strongly suggests that for a given architecture and training style, reducing underthinking (lower UT score) is aligned with better performance. We will add this crucial discussion to the paper.
>
> | Models | Accuracy (Pass@1) (↑) | UT Score (↓) |
> | --- | --- | --- |
> | Qwen3-4B | 65.6% | 15.0 |
> | Qwen3-8B | 64.6% | 16.8 |
> | Qwen3-14B | 70.8% | 13.4 |
> | Qwen3-32B | 73.8% | 10.6 |
>
> ---
>
> > Q3: The TIP decoding strategy requires a predefined set of thought-switching tokens. How are these tokens selected? What if a word is tokenized into multiple tokens?
> >
>
> A3: That's a great practical question.
>
> 1. **Selection:** The tokens (e.g., `alternatively`, `messy`) were selected empirically by observing common phrases that preceded a strategy switch in the models' outputs. While they function as tunable hyperparameters, we found these general terms worked well across tasks without extensive tuning.
> 2. **Tokenization:** When a switching keyword is split into multiple subwords, our implementation penalizes **only the first subword token** of that keyword. This ensures the penalty is applied once per intended switch.
>
> We will clarify these implementation details in the paper.
>
> ---
>
> > Q4: The legend for Figure 5 is confusing. There are dots in the figure that look different from the rest. What is the difference between these dots and the others?
> >
>
> A4: Thank you for pointing this out, and we apologize for the confusion. The dots in Figure 5 represent individual thoughts, plotting their position in the reasoning chain against the probability of being abandoned. **The visually distinct dots are simply statistical outliers and do not represent a different category of data.** They do not change the overall trend shown by the regression line. We will revise the figure caption in the final version to explicitly state this and improve clarity for the reader.

---

### Official Review · Reviewer_Ffc3 · 2025-06-30

**Clarity:** 3
**Significance:** 3
**Originality:** 3
**Rating:** 4
**Confidence:** 3

**Summary:**

The paper investigates a phenomenon called “underthinking” in long reasoning models (LRMs)—a tendency to prematurely switch between different lines of reasoning rather than fully developing promising thoughts that could lead to correct answers. The authors observe that frequent thought-switching correlates with incorrect responses across three challenging mathematics benchmarks (MATH500-Hard, GPQA-Diamond, AIME 2024) and two open-source LRMs (QwQ-32B-Preview and DeepSeek-R1). Notably, many incorrect solutions still contain at least one early correct insight, which the models fail to pursue, instead expending large numbers of tokens on unproductive reasoning paths. To quantify this behavior, the authors introduce a novel metric called the underthinking score. To mitigate the issue, they propose a decoding strategy called TIP (Thought-switching Penalty), which requires no fine-tuning and consistently improves performance across the benchmarks.

**Questions:**

Please provide some details about generalization to new domains.

**Ethical Concerns:**

["NO or VERY MINOR ethics concerns only"]

**Limitations:**

Yes the authors have discussed some limitations.

**Paper Formatting Concerns:**

No concerns.

**Quality:**

3

**Strengths And Weaknesses:**

1) Strengths

1a) Frequent Thought Switching in LRMs: The paper begins with an insightful observation: long reasoning models (LRMs) often switch between multiple solution strategies without fully committing to promising ones, even when such paths could lead to correct answers. This erratic behavior results in significant computational waste—incorrect answers tend to consume far more tokens than correct ones due to excessive idea-switching. This sheds new light on inefficiencies in tasks that demand long-term reasoning.

1b) Underthinking Metric: To quantify this behavior, the authors introduce the underthinking score—a simple yet effective ratio that captures how many tokens are used after the first correct thought in failed solutions. This metric complements traditional accuracy by offering a way to measure reasoning inefficiency across models and datasets, enabling more fine-grained evaluations and tracking of future improvements.

1c) TIP Decoding for Efficiency: Motivated by these findings, the authors propose TIP (Thought-switching Penalty) decoding, a method that discourages frequent shifts in reasoning. TIP requires no retraining and not only cuts token usage by more than half but also improves accuracy. Given the high computational costs of LRM-based reasoning, this offers a practical and impactful advance for improving efficiency and reducing latency.

1d) Effective Qualitative Insights: In addition to quantitative results, the paper includes compelling qualitative examples that clearly illustrate the underthinking phenomenon. These case studies reinforce the paper’s central claims and enhance its overall effectiveness and clarity.

2) Weaknesses

2a) Limited Model-Size Coverage: The paper’s findings are based on two very large LRMs—QwQ-32B-Preview and DeepSeek-R1-671B (plus a preview variant)—tested on three math-focused benchmarks. However, the study does not explore whether the observed “underthinking” behavior persists in smaller models from the same families or in other widely used model series (e.g., LLaMA 2 models ranging from 7B to 70B). Including a more model sizes would help assess whether underthinking is a general phenomenon or one that primarily emerges at large scales.

2b) Restricted Evaluation Scope: All experiments are limited to math reasoning and GPQA, with no investigation into whether the findings, the proposed underthinking metric, or the TIP decoding method generalize to other domains such as commonsense reasoning, dialogue, code generation, or multimodal tasks. Expanding the evaluation to diverse reasoning settings would significantly strengthen the broader applicability of the approach.

2c) Evaluation Pipeline Limitations: The underthinking score relies on an auxiliary LLM-based classifier to determine whether intermediate thoughts are “correct.” However, this classifier achieves only ~82% accuracy on a small validation set, introducing uncertainty into the core metric. Increasing the size and robustness of this evaluation could enhance the credibility of the results and reduce measurement noise.

---

> ### Author Rebuttal · Authors · 2025-07-30
>
> Thank you for your encouraging review (rating 4) and for emphasizing strengths like our insightful observation of underthinking, the novel UT metric, TIP's practical efficiency gains, and our qualitative examples. We value your suggestions for broader coverage — they've pushed us to expand our analysis.
>
> ---
>
> > Q1: Limited Model-Size Coverage
> >
>
> A1: This is an excellent point. To investigate whether underthinking is specific to very large models, we ran new experiments on a range of smaller models from the Qwen3 family (4B, 8B, 14B, 32B) on the AIME24 benchmark.
>
> Our findings are twofold and very insightful:
>
> 1. **Underthinking is a general phenomenon:** It occurs consistently across all tested model sizes, not just the largest ones.
> 2. **Underthinking severity correlates with model scale:** Within the Qwen3 family, smaller models exhibit *more* severe underthinking than their larger counterparts. For instance, the 4B model has a much higher UT Score than the 32B model.
>
> This new analysis significantly strengthens our paper by demonstrating the broad relevance of our work across model scales. We will add a new section with these results.
>
> | Models | Accuracy (Pass@1) (↑) | UT Score (↓) |
> | --- | --- | --- |
> | Qwen3-4B | 65.6% | 15.0 |
> | Qwen3-8B | 64.6% | 16.8 |
> | Qwen3-14B | 70.8% | 13.4 |
> | Qwen3-32B | 73.8% | 10.6 |
>
> ---
>
> > Q2: Restricted Evaluation Scope
> >
>
> A2: Thank you for the valuable suggestion. Inspired by your feedback, we conducted new experiments on a multimodal benchmark.
>
> **Our analysis toolkit generalizes well to multimodal reasoning.** We evaluated two multimodal LRMs (QVQ-72B-Preview, GLM-4.1V-Thinking) on the OE_MM_maths_en_COMP subset of OlympiadBench, which requires reasoning over both text and images. The results clearly show that the underthinking phenomenon persists in this more complex, multimodal setting.
>
> | Models | Accuracy (↑) | UT Score (↓) |
> | --- | --- | --- |
> | QVQ-72B-Preview | 46.0% | 33.6 |
> | GLM-4.1V-Thinking | 58.7% | 17.1 |
>
> This confirms that underthinking is a broad challenge not limited to text-only math problems. We will add these new results and a discussion to the final manuscript to strengthen the paper's general applicability.

---

### Official Review · Reviewer_4W6S · 2025-07-03

**Clarity:** 3
**Significance:** 3
**Originality:** 3
**Rating:** 4
**Confidence:** 4

**Summary:**

This work investigates the phenomenon of underthinking in current LRMs, where models frequently switch to different reasoning thoughts without fully exploring the path. It designs token efficiency metrics to measure such behaviors in QwQ-32B-Preview and Deepseek-R1-671B. To mitigate underthinking issues, this work develops Thought Switching Penalty to decrease the logit values during inference. The approach is evaluated on AIME-24 demonstrating that TIP reduces underthinking and improves accuracy .

**Questions:**

1. The TIP method works by penalizing the logits of certain keywords (like “alternatively”) that indicate thought switching. But some of these keywords might also reflect other useful cognitive behaviors, as discussed in a related work [1]. Do you think TIP might unintentionally suppress those behaviors too? Any insight or analysis would be helpful.

[1] Cognitive Behaviors that Enable Self-Improving Reasoners, or, Four Habits of Highly Effective STaRs.

**Ethical Concerns:**

["NO or VERY MINOR ethics concerns only"]

**Final Justification:**

I have carefully read the authors' responses and encourage the authors to incorporate these results and clarifications into the revision. I've noted that some of the clarifications rely on hypotheses without concrete evidence, which is the main reason I am hesitant to provide a higher score. From the empirical side this paper makes a timely and valuable contribution.

**Limitations:**

The authors briefly mention limitations in Section 2.3, but a separate section outlining them more clearly would be helpful.

**Quality:**

3

**Strengths And Weaknesses:**

Strengths:

1. This paper is well-written and content is clear and straightforward.
2. The underthinking is an interesting concept to study the reasoning behaviors, providing a useful perspective in understanding LRMs.
3. The work presents a thorough analysis of the correctness of intermediate thoughts and provides convincing empirical evidence to validate the presence of underthinking.

Weaknesses:
1. Table 2 is missing results for R1-Distill-Qwen-32B and Deepseek-R1 on MATH500-hard and GPQA Diamond. Also, from the table,  the results suggest that TIP yields more substantial accuracy improvements for models with lower baseline performance (such as QwQ on GPQA and AIME-24). Do you have any insight on this?

2. (minor) The paper does not clearly explain why underthinking behavior appears in LRMs. A further discussion of its possible causes would help improve understanding.

---

> ### Author Rebuttal · Authors · 2025-07-30
>
> Thank you for your supportive review and for praising the paper's clarity, the interesting concept of underthinking as a lens on LRMs, and our thorough analysis with convincing evidence. We're grateful for your detailed questions, which helped us refine our insights.
>
> ---
>
> > Q1: Table 2 is missing results for R1-Distill-Qwen-32B and Deepseek-R1 on MATH500-hard and GPQA Diamond. Also, from the table, the results suggest that TIP yields more substantial accuracy improvements for models with lower baseline performance (such as QwQ on GPQA and AIME-24). Do you have any insight on this?
> >
>
> A1: Thank you for pointing this out. We have updated Table 2 with new results.
>
> 1. **New Results:** We have added results for **R1-Distill-Qwen-32B** on MATH500-hard and GPQA Diamond. These results reinforce our claims: the model exhibits underthinking, and our TIP method successfully mitigates it to improve accuracy. Due to the limited rebuttal period and high computational cost, we were unable to complete the runs for DeepSeek-R1 but will ensure they are included in the final camera-ready version.
> 2. **Insight on Performance Gains:** That’s a sharp observation. **We hypothesize that models with lower baseline accuracy (like QwQ on GPQA) may have been trained on data that inadvertently encourages underthinking, or were simply less optimized against this specific failure mode.** Consequently, TIP, which directly targets this issue, provides a larger relative boost. However, we emphasize that TIP consistently improves performance across *all* tested models, demonstrating its broad utility.
>
> | Models | **Accuracy (Pass@1) (↑)** | **Thought Number**| **Thought Interval**|**Weighted UT Score (↓)** |
> | --- | --- | --- | --- | --- |
> | **MATH500-Hard (Level 5)** |  |  |  |  |
> | QwQ-32B-Preview | 83.1 | 12.6 | 445.6 | 11.7±20.5 |
> | + TIP | 83.7 | 5.7 | 517.6 | 11.0±19.5 |
> | R1-Distill-Qwen-32B | 88.3 | 6.7 | 792.5 | 3.3±8.8 |
> | + TIP | 89.4 | 2.7 | 964.0 | 3.0±8.5 |
> | **GPQA Diamond** |  |  |  |  |
> | QwQ-32B-Preview | 57.6 | 21.1 | 356.8 | 25.1±23.9 |
> | + TIP | 59.1 | 7.3 | 432.5 | 23.2±23.2 |
> | R1-Distill-Qwen-32B | 61.6 | 13.4 | 548.6 | 22.3±25.0 |
> | + TIP | 61.7 | 4.6 | 747.1 | 23.1±25.3 |
> | **AIME2024** |  |  |  |  |
> | QwQ-32B-Preview | 38.3 | 16.1 | 459.7 | 40.6±28.4 |
> | + TIP | 44.1 | 13.9 | 515.7 | 35.8±27.8 |
> | R1-Distill-Qwen-32B | 61.4 | 8.2 | 819.5 | 19.6±20.6 |
> | + TIP | 64.1 | 4.5 | 1018.0 | 17.7±20.6 |
> | DeepSeek-R1 | 73.8 | 13.8 | 580.1 | 14.6±19.1 |
> | + TIP | 74.8 | 5.7 | 941.6 | 13.0±18.0 |
>
> ---
>
> > Q2: The paper does not clearly explain why underthinking behavior appears in LRMs. A further discussion of its possible causes would help improve understanding.
> >
>
> A2: Thank you for asking for clarification. We hypothesize that underthinking is an unintended byproduct of training LRMs to mimic complex, multi-path human reasoning. The causal chain is likely:
>
> 1. To enable deep reasoning, models like OpenAI's o1 demonstrated the value of exploring multiple reasoning paths.
> 2. This influenced the creation of supervised fine-tuning (SFT) datasets, which likely over-represented examples with frequent thought-switching to encourage this exploratory behavior.
> 3. As a result, models learn to switch paths too readily, even when a current path is promising. This leads to the premature abandonment of good ideas that we term "underthinking".
>
> We will add this explicit explanation to the paper to provide better context.
>
> ---
>
> > Q3: The TIP method works by penalizing the logits of certain keywords (like “alternatively”) that indicate thought switching. But some of these keywords might also reflect other useful cognitive behaviors, as discussed in a related work. Do you think TIP might unintentionally suppress those behaviors too? Any insight or analysis would be helpful.
> >
>
> A3: That's a great question. **No, we designed TIP specifically to avoid this.** TIP does not penalize all thought-switching, only behavior indicative of *underthinking*: frequent, rapid, and unproductive switching.
> Specifically, TIP applies a penalty only when a switching keyword (e.g., "alternatively") appears with high frequency in a short span (in our work, ≥2 times within 600 tokens). An isolated, deliberate switch to a new strategy (like backtracking or reflection, as in [1]) would not be penalized. Therefore, TIP complements, rather than suppresses, productive reasoning strategies. We will clarify this important nuance in the revised manuscript.
>
> [1] Cognitive Behaviors that Enable Self-Improving Reasoners, or, Four Habits of Highly Effective STaRs.

---

> > ### Comment · Reviewer_4W6S · 2025-08-07
> > **Thank you for the response**
> >
> > Thank you for providing the additional experiments and insights. I have carefully read the authors' responses and found the clarifications of Q3 to be especially helpful and encourage the authors to incorporate these results and clarifications into the revision. I am pleased to maintain my positive judgment.

---

> > > ### Author Response · Authors · 2025-08-07
> > >
> > > Thank you for your positive feedback and valuable suggestions. We greatly appreciate your time and effort, and we will incorporate all clarifications regarding Q3 into the revised manuscript.​

---

### Official Review · Reviewer_Po39 · 2025-07-03

**Clarity:** 3
**Significance:** 3
**Originality:** 3
**Rating:** 5
**Confidence:** 4

**Summary:**

The paper identifies underthinking in long-reasoning LLMs (LRMs)—a tendency to hop between reasoning threads without sufficiently exploring promising ones. It (i) formalises an underthinking score that captures token-efficiency in incorrect answers
, (ii) shows the prevalence of the phenomenon across three maths-reasoning benchmarks, and (iii) proposes Thought-switching-penalty (TIP), a decoding modification that down-weights tokens signalling a strategy switch. TIP improves Pass@1 by up to +5.8 pp without fine-tuning and synergises with self-consistency and laconic decoding

**Questions:**

See the weakness part.

**Ethical Concerns:**

["NO or VERY MINOR ethics concerns only"]

**Final Justification:**

I will maintain the original score, in my opinion this paper should be accepted.

**Limitations:**

yes.

**Quality:**

3

**Strengths And Weaknesses:**

Strength

1. To the best of my knowledge, this is the first paper that systematically characterise the "underthinking" in reasoning LLMs.

2. Therefore this is also the first paper that proposed the SUT score, a valid metric for quantifying the underthinking behaviours.

3. This paper really expanded my understanding of the LLMs' limitations beyond the simple accuracy metrics.

Weakness

1. The correct thought score is judged by the distilled LLMs, which may introduce bias.

2. Rather than saying as a weakness point, I am a little bit curious that if this kind of analyse toolkit could be easily adapt to coding and multi-modal reasoning problems?

---

> ### Author Rebuttal · Authors · 2025-07-30
>
> Thank you for your positive assessment (rating 5) and for highlighting the novelty of our systematic characterization of underthinking in LRMs, our proposed SUT score (which you called a "valid metric"), and how it expands understanding beyond simple accuracy metrics. We appreciate your curiosity about extensions and are glad our work resonated with you.
>
> ---
> > Q1: The correct thought score is judged by the distilled LLMs, which may introduce bias.
> >
>
> A1: We agree this is a critical aspect and have already quantified the classifier's reliability in the paper. As detailed in Lines 131-137, we validated our LLM-based classifier on a manually annotated set. It achieves **82.9% accuracy on correct thoughts and 81.8% on incorrect thoughts.** While not perfect, this level of accuracy is sufficiently high to reliably identify the trends central to our paper's claims, such as the correlation between frequent thought-switching and incorrect final answers.
>
> We believe this establishes a strong baseline for this novel analysis task. We fully agree that developing more robust, model-independent evaluation methods is a valuable direction for future work, and we will emphasize this in the limitations section of the revised paper.
>
> ---
>
> > Q2: Rather than saying as a weakness point, I am a little bit curious that if this kind of analyse toolkit could be easily adapt to coding and multi-modal reasoning problems?
> >
>
> A2: Thanks for suggesting we explore the applicability of our findings beyond math reasoning. Inspired by your feedback, we conducted new experiments on a multimodal benchmark.
>
> **Our analysis toolkit generalizes well to multimodal reasoning.** We evaluated two multimodal LRMs (QVQ-72B-Preview, GLM-4.1V-Thinking) on the OE_MM_maths_en_COMP subset of OlympiadBench, which requires reasoning over both text and images. The results clearly show that the underthinking phenomenon persists in this more complex, multimodal setting.
>
> | Models | Accuracy (↑) | UT Score (↓) |
> | --- | --- | --- |
> | QVQ-72B-Preview | 46.0% | 33.6 |
> | GLM-4.1V-Thinking | 58.7% | 17.1 |
>
> This confirms that underthinking is a broad challenge not limited to text-only math problems. We will add these new results and a discussion to the final manuscript to strengthen the paper's general applicability.

---

### Decision · Program_Chairs · 2025-09-17

**Decision:**

Accept (spotlight)

**Comment:**

This paper proposes a novel decoding strategy and provide experimental results to demonstrates that the proposed method can improve accuracy across challenging datasets without requiring model fine-tuning. The paper is well written and the proposed idea is innovative. However, the analyses done to evaluate the technique are questionable (scoring, further discussion of causes, model-size coverage, limitations). It is suggested that the authors further revise the paper and address the issues raised by the reviewers. We recommend accepting this paper.